# PHYSICS-AWARE TENSOR FIELD NEURAL PDE FOR CLIMATE AND WEATHER PREDICTION

## ABSTRACT

Climate and weather prediction has traditionally relied on computationally demanding numerical simulations grounded in atmospheric physics, yet deep-learning approaches are emerging as transformative alternatives. Existing methods, however, are often purely data-driven and physics-agnostic, overlooking essential physical principles and struggling to generalize. To address these challenges, we present the Physics-Aware Tensor Field Neural PDE (PA-TFNP), a forecasting framework that embeds rotation-equivariant tensor-field neural operators directly on the sphere, couples them with a numerically rigorous gradient operator based on spherical transforms and physically consistent boundary treatment, and augments the learned dynamics with diffusion terms derived from the atmospheric primitive equations. These innovations enable our model to achieve superior performance through strict physical fidelity and efficient learning. The proposed PA-TFNP achieves state-of-the-art performance in global and regional weather prediction, outperforming ClimODE by 78.92% on global hourly data with a comparable number of parameters.

## 1 INTRODUCTION

Accurate climate and weather prediction is crucial for understanding environmental phenomena, preparing for extreme events, and enabling informed decisions. Traditional numerical simulations grounded in atmospheric physics (Rabier et al., 2000; Rawlins et al., 2007; Thompson, 1961) have achieved remarkable accuracy over medium timescales, leveraging systems of partial differential equations (PDEs) to model atmospheric dynamics and capture processes like advection, diffusion, and thermodynamics (Lions et al., 1992; Haltiner, 1971; Coiffier, 2011). However, solving these PDEs is computationally expensive, and extensive or proprietary datasets (Yu, 2010; Warner, 2010) pose significant scalability challenges, often making real-time or high-resolution global predictions infeasible. Moreover, traditional models struggle with rapidly changing climate patterns not well-represented in historical data (Neelin, 2010), highlighting the need for methods that are computationally efficient and can learn from observed data while maintaining physical consistency (Bader et al., 2008).

In recent years, machine learning approaches have emerged as transformative alternatives to traditional simulations, challenging the mechanistic modeling paradigm with data-driven methods (Bi et al., 2023; Lam et al., 2023; Bodnar et al., 2024; Kochkov et al., 2024). These models learn complex spatiotemporal patterns directly from observations, bypassing the need to solve costly PDEs. They have shown promise in tasks ranging from high-resolution weather forecasting to global climate simulations (Bihlo, 2021; Verma et al., 2024; Pathak et al., 2022), capturing intricate dependencies for near-term predictions and localized events. Despite these successes, many remain physics-agnostic, relying solely on learned correlations rather than leveraging physical principles. Consequently, they struggle to enforce fundamental conservation laws, such as mass or energy conservation, and lack mechanisms to maintain incompressibility in fluid dynamics. This limits their generalization across diverse geophysical scenarios and leads to error accumulation over extended timeframes, undermining long-term forecasting reliability. To address these limitations, we propose the Physics-Aware Tensor Field Neural PDE (PA-TFNP), a novel framework designed to enhance climate and weather prediction by combining the strengths of deep learning with physical principles. In contrast to recent neural surrogates—such as ClimODE and ClimaX—that operate on flattened latitude–longitude grids or impose physics only through auxiliary losses, PA-TFNP learns directly on spherical tensor fields, preserving rotational symmetry throughout the network. It fuses a rotation-equivariant tensor-

field operator with a mathematically consistent spherical-transform gradient and physically sound boundary conditions, giving the model intrinsic knowledge of physics laws rather than relying on post-hoc corrections. Furthermore, PA-TFNP embeds diffusion dynamics explicitly derived from the atmospheric primitive equations, enabling realistic long-term dynamics. This integration of geometry, numerics, and physics delivers substantial gains over existing benchmarks while demanding significantly fewer computational resources, proving that physical fidelity and efficiency can coexist in modern weather-forecasting systems. Our key contributions are as follows:

- We propose a Tensor Field Neural PDE framework (TFNP) powered by tensor-field neural networks that not only captures rotationally equivariant spatiotemporal patterns but also consistently outperforms the latest benchmark models across diverse climate and weather-prediction tasks.

- We devise a numerically rigorous spherical-transform-based gradient operator with physically consistent boundary conditions that stabilizes training and sharpens predictive precision, particularly near domain boundaries.

- We embed diffusion dynamics informed by the atmospheric Primitive Equations into our network, capturing key atmospheric processes and thereby improving both the accuracy and stability of weather forecasts.

Through these contributions, our method achieves significant improvements in both accuracy and robustness, effectively bridging the gap between physics-driven simulations and data-driven machine learning approaches.

## 2 RELATED WORKS

**Numerical weather prediction**. Conventional climate and weather forecasting primarily depends on physics-based numerical simulations (Shuman, 1989; Warner, 2010). In particular, short-term forecasts rely on established Numerical Weather Prediction (NWP) systems—such as the Unified Model (UM) (Bush et al., 2020) or other frameworks used in the U.S. (Powers et al., 2017) and Europe—that solve the so-called primitive equations (Wedi et al., 2015), a topic of extensive mathematical and computational research (Lions et al., 1992). Meanwhile, longer-term forecasts employ dedicated climate models, with Earth System Models (ESMs) (Mukhopadhyay et al., 2019) representing the cutting edge by coupling atmospheric, cryospheric, terrestrial, and oceanic processes. Although these modeling approaches have seen considerable success, they still face notable challenges, including sensitivity to initial conditions, structural inconsistencies across models (Bauer et al., 2015), significant computational burdens, and marked regional variability.

**Deep learning for forecasting**. Recent advances in deep learning have yielded promising results for weather forecasting by bypassing some of the complexities of physics-based simulations. For instance, Rasp et al. (2020) applied pre-training with ResNet for medium-range weather prediction, and utilized large ensembles of deep models to capture sub-seasonal variations (Han et al., 2024). Other notable works include radar-based deep generative models for nowcasting (Ravuri et al., 2021) and graph neural network-based forecasting in GraphCast (Lam et al., 2023). In addition, FourCastNet (Kurth et al., 2023) and Pangu-Weather (Bi et al., 2023) represent state-of-the-art neural forecasting approaches that harness data-driven backbones, such as Vision Transformer, UNet, and autoencoders. Despite their empirical strengths, these methods tend to overlook key physical principles and seldom provide uncertainty estimates, limiting their interpretability and robustness.

**Physics-Informed Machine Learning**. Neural ODEs frame time derivatives as learnable neural networks (Fermanian et al., 2021), and have been extended to incorporate physics-based constraints (Verma et al., 2024). Physics-Informed Neural Networks (PINNs) (Cai et al., 2021) embed mechanistic knowledge into DEs, and a broader line of research focuses on discovering interpretable differential equations (Brunton and Kutz, 2024). Extending such ideas to Neural PDEs often requires specialized spatial discretizations (Kochkov et al., 2024) or functional representations (Seol et al., 2024). Several studies have also used machine learning to improve fluid dynamics models (Choi et al., 2024). Notably, most of these works deal with smaller-scale fluid systems rather than the global scope demanded by climate or weather applications.

## 3 METHODOLOGY

Our model is fundamentally constructed using the Method of Lines (MOL) framework, as described in (Verma et al., 2024). This approach initially formulates the problem in terms of partial differential equations (PDEs) governing the evolution of multiple variables. To approximate the spatial derivatives in these PDEs, we employ a finite difference scheme, converting the PDEs into a system of ordinary differential equations (ODEs). Subsequently, we effectively approximate the temporal dynamics of the atmospheric variables by solving this system through a neural ODE framework (Chen et al., 2018). The detailed formulation is outlined below.

### 3.1 PRELIMINARY

Consider a set of $d$ atmospheric variables denoted by $\mathbf{q}(\mathbf{x}, t) = \{q_i(\mathbf{x}, t)\}_{i=1}^d$ (e.g. temperature, geopotential height) that depend on the spatial location $\mathbf{x} \in [-90, 90] \times [0, 360]$ (representing latitude and longitude on a sphere domain, such as Earth) and time $t > 0$. Observations of these variables are collected at a set of uniform grid points $\{\mathbf{x}_n\}_{n=1}^N$, where the spatial domain consists of $H$ latitude points and $W$ longitude points, resulting in a total of $N = HW$ observations. In addition, we can consider the velocity field $\mathbf{U}(t) = \{\{\mathbf{u}_i(\mathbf{x}_n, t)\}_{i=1}^d\}_{n=1}^N$ that governs the advection of atmospheric variables. Given the velocity field, we model the temporal evolution of these variables using the following governing equations as in (Verma et al., 2024).

$$\frac{\partial}{\partial t} q_i(\mathbf{x}, t) = -\mathbf{u}_i(\mathbf{x}, t) \cdot \nabla q_i(\mathbf{x}, t) - q_i(\mathbf{x}, t) \nabla \cdot \mathbf{u}_i(\mathbf{x}, t), \tag{1}$$

$$\frac{\partial}{\partial t} \mathbf{u}_i(\mathbf{x}, t) = f_\eta \left( \mathbf{Q}(t), \nabla \mathbf{Q}(t), \mathbf{U}(t), g(\{\mathbf{x}_n\}_{n=1}^N, t) \right),$$

where $\nabla$ denotes the spatial gradient, $\mathbf{Q}(t)$ represents the set $\{\mathbf{q}(\mathbf{x}_n, t)\}_{n=1}^N$, $g$ is a spatio-temporal embedding function and $f_\eta$ is a trainable neural network with parameter $\eta$. The second equation implies that the velocity of each variable could be influenced by the other variables.

To transform Equation (1) into a system of ODEs, we approximate the spatial derivatives using a finite difference, denoted as $\widehat{F}$ (see Section 3.3 for details). The system for all variables $Q(t)$ at the points of the grid is given below.

$$\frac{d\mathbf{Q}(t)}{dt} = \begin{pmatrix} \frac{\partial \mathbf{q}(\mathbf{x}_1, t)}{\partial t} \\ \vdots \\ \frac{\partial \mathbf{q}(\mathbf{x}_N, t)}{\partial t} \end{pmatrix} \approx \begin{pmatrix} \widehat{F}(\mathbf{q}(\mathbf{x}_1, t), \{\mathbf{q}(\mathbf{x}_n, t)\}_{n \in \mathcal{N}(1)}), \mathbf{u}(\mathbf{x}_1, t), \{\mathbf{u}(\mathbf{x}_n, t)\}_{n \in \mathcal{N}(1)}) \\ \vdots \\ \widehat{F}(\mathbf{q}(\mathbf{x}_N, t), \{\mathbf{q}(\mathbf{x}_n, t)\}_{n \in \mathcal{N}(N)}), \mathbf{u}(\mathbf{x}_N, t), \{\mathbf{u}(\mathbf{x}_n, t)\}_{n \in \mathcal{N}(N)}) \end{pmatrix} \in \mathbb{R}^{Nd}.$$

Here, $\mathcal{N}(i)$ denotes the index set corresponding to the neighborhood of the grid point $\mathbf{x}_i$ required for the finite-difference approximation. The system that governs $\mathbf{U}(t)$ can be formulated analogously. Consequently, the complete system consists of $3Nd$ components when each atmospheric variable is considered a separate component. By integrating Equation (2) using the Runge-Kutta method to solve this system, we can estimate the values of the variables $\{q_i\}_{i=1}^d$ at all grid points $\{\mathbf{x}_n\}_{n=1}^N$.

$$\begin{bmatrix} \mathbf{Q}(t) \\ \mathbf{U}(t) \end{bmatrix} = \begin{bmatrix} \mathbf{Q}(t_0) \\ \mathbf{U}(t_0) \end{bmatrix} + \int_{t_0}^t \begin{pmatrix} \frac{d\mathbf{Q}(s)}{ds} \\ \frac{d\mathbf{U}(s)}{ds} \end{pmatrix} ds \tag{2}$$

Using the estimated $\mathbf{Q}(t)$ and real data, we train $f_\eta$ by minimizing the negative log-likelihood loss function, as defined in Sections 3.7 and 3.8 of (Verma et al., 2024).

### 3.2 TENSOR FIELD NEURAL PDE (TFNP)

In this paper, we parametrize the nonlinear operator $f_\eta$ in Equation (1) (illustrated in Figure 1) with a Tensor Field Network (TFN) $f_{TFN}$ (Thomas et al., 2018; Weiler et al., 2018; Kondor et al., 2018), combined with an attention mechanism, $f_{att}$ (Vaswani et al., 2017), rather than employing a convolutional neural network (CNN). Although CNNs are often adopted for $f_\eta$ because they can approximate finite difference schemes on a uniform Euclidean grid (Brandstetter et al., 2022; Long et al., 2018), global climate data are typically sampled uniformly in latitude and longitude coordinates.

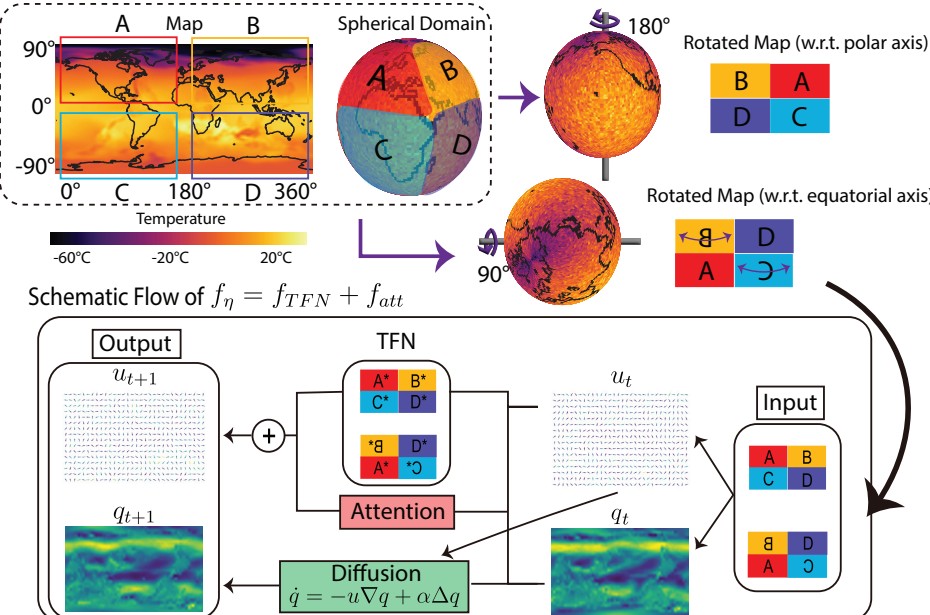

Figure 1: Graphical overview of PA-TFNP. The tensor field network (TFN) and attention layer are employed to model $f_\eta$ and the advection-diffusion equation is introduced. TFN accounts for the spherical geometry of Earth. For instance, earth can be divided into four regions (A, B, C, D) based on latitude $[0°, 90°]$, $[-90°, 0°]$ and longitude $[0°, 180°]$, $[180°, 360°]$. Projecting temperature data onto the latitude-longitude plane forms the leftmost rectangular map. Rotation around the polar axis leads to translation on this map, while rotation around the equatorial axis additionally reflects region B and C. PA-TFNP processes these partitioned region and outputs corresponding (A*, B*, C*, D*), ensuring rotational equivariance. Combining PA-TFNP with attention yields the final model $f_\eta$.

This leads to geometric distortions near the polar regions, negatively impacting prediction accuracy. Moreover, CNNs inherently fail to capture rotation-equivariant properties essential for processing spherical data. As in Figure 1, while rotations around the polar axis correspond to straightforward transformations in a periodic domain, rotations around the equatorial axis involve transformations coupled with reflections. Consequently, a CNN with fixed filters cannot approximate rotations of the latter type, as local features along the boundaries separating regions $A$, $B$, $C$, and $D$ become distorted. We adopted a neural network based on tensor products instead of CNNs to mitigate this problem. This approach is inherently rotation equivariant, ensuring that transformations affect points near the poles and the equator consistently, without introducing distortion. The detailed formulation is as follows. The function $f_\eta$ takes as input $\mathbf{Q}(t) \in \mathbb{R}^{N \times d}$, $\nabla \mathbf{Q}(t) \in \mathbb{R}^{N \times 2d}$, $\mathbf{U}(t) \in \mathbb{R}^{N \times 2d}$, and $g(\{x_n\}_{n=1}^N, t) \in \mathbb{R}^{N \times e}$, where $e$ denotes the embedding dimension introduced by $g$. If inputs in $T$ time steps are considered simultaneously, the dimension of input $I$ is given by $T \times N \times (5d+e)$. After reshaping $I$ into a tensor of size $N \times C_{in}$, we can define the neural network $f_\eta : \mathbb{R}^{N \times C_{in}} \to \mathbb{R}^{N \times C_{out}}$ as a tensor product-based function. This function is parameterized by a trainable weight tensor $W[c_{out}, c_1.c_2]$ for indices $c_{out}, c_1, c_2 \in [C_{out}], [C_{in}], [C_{in}]$, and is formulated as:

$$f_{TFN}(I[i, c_{\text{out}}]) = I \otimes I = \sum_{c_1=1}^{C_{\text{out}}} \sum_{c_2=1}^{C_{\text{out}}} W[c_{\text{out}}, c_1, c_2](I[i, c_1] \cdot I[i, c_2]), \quad \forall i \in [N].$$

Here, $C_{in}, C_{out}$ denote the input and output channel dimensions of $f_{TFN}$. Additionally, we incorporate an attention-based network, $f_{att}$, following the architecture proposed in (Verma et al., 2024). Consequently, the final $f_\eta$ is constructed as the sum of the attention network $f_{att}$ and the Tensor Field Network $f_{TFN}$,

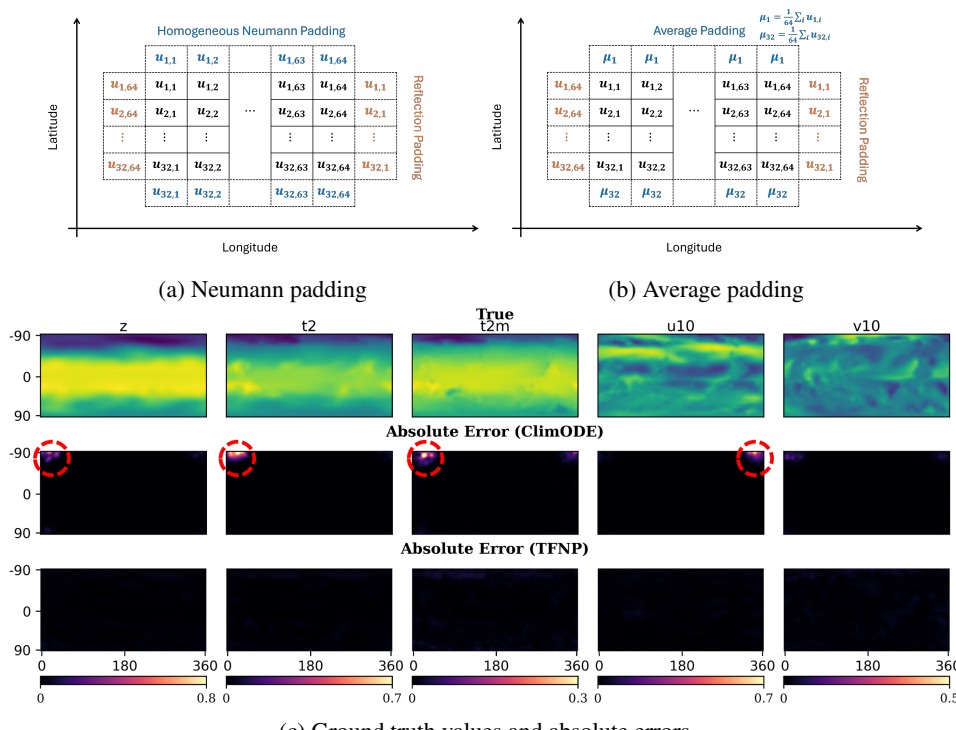

(a) Neumann padding

(b) Average padding

(c) Ground truth values and absolute errors

Figure 2: (a) Description of Neumann padding. (b) Description of the average padding. (c) Ground truth values for z, t2, t2m, u10, and v10 and absolute errors of ClimODE and the proposed TFNP.

### 3.3 PHYSICS-AWARE TENSOR FIELD NEURAL PDE (PA-TFNP)

In this section, we introduce PA-TFNP, an extension of TFNP that incorporates physical constraints into the model. We make three key modifications. First, we apply boundary conditions that reflect the domain's physical properties. Second, spatial derivatives are computed using spherical operators to capture Earth's geometry. Third, we augment the inputs to $f_\eta(\cdot)$ in Equation equation 1 with physically relevant features: ground wind magnitude, lapse rate, and wind vorticity. We also modify the PDE solver to blend neural outputs with physics-based tendencies for improved interpretability and fidelity.

#### BOUNDARY CONDITIONS

ClimODE exhibits unexpected errors near the boundary of the domain (see Figure 2), primarily due to the discretization of the sphere onto a longitude–latitude rectangular domain. This issue arises from the absence of proper boundary conditions in the original ClimODE formulation (Verma et al., 2024). The boundary conditions are implemented through an appropriate padding strategy and incorporated into the advection–diffusion equation during gradient computation. We propose two padding strategies, Neumann padding and average padding, both reflecting the physical characteristics of the domain.

In both strategies, circular padding is applied along the longitudinal boundaries, effectively transforming the rectangular domain into a cylindrical one. For Neumann padding, replicate padding is used along the latitudinal boundaries, corresponding to homogeneous Neumann boundary conditions at the north and south poles (see Figure 2a). In the case of average padding, we extend the domain by padding with the average values of the boundary: $\mu_1 = \frac{1}{64} \sum_{i=1}^{64} u_{1,i}$ and $\mu_2 = \frac{1}{64} \sum_{i=1}^{64} u_{2,i}$. This transforms the rectangular domain into a sphere-like domain (see Figure 2b). Figure 2c illustrates that TFN, equipped with this padding scheme, effectively captures the solution behavior near the poles. With a rotation-equivariant property, TFNP maintains consistent prediction accuracy across

all regions, including areas near the poles, resulting in robust predictions both at the boundaries and within the domain interior.

### SPATIAL DERIVATIVE APPROXIMATION

This section outlines the computation of the spatial derivatives in Equation (1). The method proposed in (Verma et al., 2024) estimates the derivatives by directly computing finite difference approximations along latitude and longitude, respectively. However, in a spherical domain, a given longitudinal difference corresponds to varying Euclidean distances depending on latitude. To account for this, we adopt a central finite difference scheme with a distance correction term:

$$\nabla q_i((\phi, \lambda), t)$$
$$\approx \left( \frac{q_i((\phi + h, \lambda), t) - q_i((\phi - h, \lambda), t)}{Rh\pi/180}, \frac{q_i((\phi, \lambda + w), t) - q_i((\phi, \lambda - w), t)}{Rh\pi \cos\phi/180} \right), \tag{3}$$

where $R$ represents the Earth's radius, and $h$ and $w$ denote the uniform grid spacing in latitude and longitude, respectively. Given the inherent periodicity in the longitudinal direction ($\lambda$), all grid points along this axis can be treated as interior points. Furthermore, we impose boundary conditions such as Neumann or periodic conditions on the latitude ($\phi$), ensuring that all points within the domain are treated as interior points. Under these conditions, the central finite difference scheme can be consistently applied throughout the entire domain.

### ADDITIONAL PHYSICS-DERIVED FEATURES

To augment the original TFNP framework, we introduce three physics-informed features: **(i)** the near-surface wind magnitude $|\boldsymbol{V}_{10}| = \sqrt{u_{10}^2 + v_{10}^2}$, **(ii)** the low-tropospheric lapse rate $\Delta t = t - t_{2m}$, and **(iii)** the relative vorticity $\zeta = \partial_y v_{10} - \partial_x u_{10}$, computed using spherical gradients. These quantities capture dynamic and thermodynamic processes essential to atmospheric motion.

### MODIFIED PRIMITIVE EQUATION

To improve physical realism and long-term stability, we extend the neural advection formulation in Equation 1 by incorporating physics-inspired diffusion and momentum correction terms.

First, scalar quantities such as temperature, humidity, and geopotential exhibit diffusive behavior in the real atmosphere, caused by unresolved subgrid turbulence and eddy transport Haltiner (1971); Lions et al. (1992); Warner (2010). To reflect this, we introduce a spatially varying diffusion term with a learnable non-negative coefficient $\alpha(\mathbf{x}) \in \mathbb{R}^{d \times H \times W}$. The scalar transport equation is modified as follows:

$$\frac{\partial q_i(\mathbf{x}, t)}{\partial t} = -\mathbf{u}_i(\mathbf{x}, t) \cdot \nabla q_i(\mathbf{x}, t) - q_i(\mathbf{x}, t) \nabla \cdot \mathbf{u}_i(\mathbf{x}, t) + \alpha(\mathbf{x}) \Delta q_i(\mathbf{x}, t),$$

where the last term mimics anisotropic and spatially varying diffusion. Next, we augment the neural tendency with physically meaningful momentum dynamics for the learned velocity field $\mathbf{u}_i$. Specifically, we apply a time-dependent blending of neural predictions and physically grounded operators:

$$\frac{\partial \mathbf{u}_i(\mathbf{x}, t)}{\partial t} = (1 - \beta_t) f_\eta \big( \mathbf{Q}(t), \nabla \mathbf{Q}(t), \mathbf{U}(t), g(\{\mathbf{x}_n\}_{n=1}^N, t) \big) + \beta_t f_{\text{phys}}(\mathbf{x}, t, \mathbf{u}_i),$$

where the blend factor $\beta_t = 1 - \exp(-t/\tau_0)$ gradually shifts preference from neural inference to physical consistency over time. The physical operator $f_{\text{phys}}$ imposes structure on the velocity evolution by incorporating key dynamical effects:

$$f_{\text{phys}}(\mathbf{x}, t, \mathbf{u}_i) = -\nabla \Phi + \nu \Delta \mathbf{u}_i - \gamma \mathbf{u}_i,$$

where $\Phi$ denotes the geopotential field (i.e., $\Phi = z$), and $\nu, \gamma$ are learnable viscosity and linear drag coefficients, respectively. This hybrid formulation preserves the expressiveness of neural models while enforcing core physical constraints, improving both predictive performance and stability in long-range forecasts.

## 4 EXPERIMENTS

We evaluate the performance of PA-TFNP by comparing it with the neural ODE, ClimaX (Nguyen et al., 2023) and ClimODE (Verma et al., 2024), a state-of-the-art data-driven global climate forecasting model. To ensure a fair comparison, we follow the experimental setup of (Verma et al., 2024), except for specific modifications detailed below. We utilize the ERA5 dataset from Weather-Bench (Rasp et al., 2020), selecting $d = 5$ key atmospheric variables: ground temperature ($t2m$), atmospheric temperature ($t$), geopotential height ($z$), and ground wind components ($u10, v10$). All variables are normalized to the range [0, 1] using min-max scaling. Further details on dataset preprocessing and training settings remain consistent with those in (Verma et al., 2024) and Appendix B. All experiments were conducted using a single RTX 4090 GPU.

### 4.1 GLOBAL WEATHER FORECASTING ACROSS VARYING TEMPORAL AND SPATIAL RESOLUTIONS

To evaluate the scalability and generalization of PA–TFNP across both spatial and temporal dimensions, we conduct experiments on global weather forecasting at two different settings: (a) long-term prediction over 5 days at a coarse resolution ($5.625°$), and (b) short-term prediction over 6 to 42 hours at a finer resolution ($11.25°$). Figure 3 summarizes the RMSE results for the five key atmospheric variables (z, t, t2m, u10, v10), comparing PA–TFNP with the state-of-the-art ClimODE baseline.

Across both resolutions, PA–TFNP consistently outperforms ClimODE. In the long-term setting (first row in Figure 3), our model demonstrates particularly large improvements in forecasting geopotential height and atmospheric temperature. Similarly, in the short-term setting (second row in Figure 3), PA–TFNP shows improved accuracy across all lead times, with gains becoming more pronounced beyond 24 hours. This indicates that the model maintains robustness even as the forecasting horizon increases. These results confirm the effectiveness of PA–TFNP in learning global-scale spatiotemporal dynamics, while preserving accuracy across varying resolutions and forecast ranges.

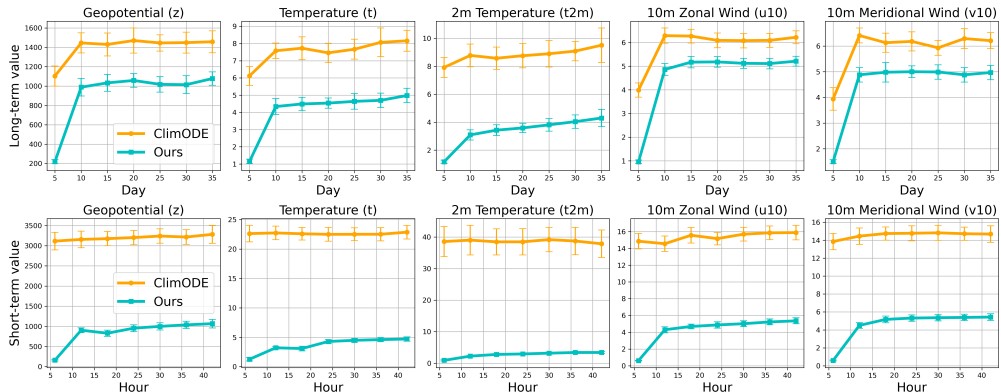

Figure 3: Comparison of RMSE values for ClimODE and the proposed PA-TFNP (Ours) across two spatiotemporal resolutions. The results highlight the performance differences for key atmospheric variables. Results are reported as mean $\pm$ standard deviation. First row: long-term prediction at a resolution of $5.625°$. Second row: Short-term prediction at a resolution of $11.25°$. PA-TFNP outperforms ClimODE by 38.12% on daily data and by 78.92% on hourly data.

### 4.2 SHORT-TERM REGIONAL WEATHER FORECASTING

We evaluate short-term (up to 24 hours) regional weather forecasting over the Australia and the South American region. Table 1 presents the RMSE (mean $\pm$ standard deviation) of various models across five key atmospheric variables. Our proposed model, PA–TFNP, demonstrates strong predictive accuracy overall, particularly for the geopotential height (z) and temperature (t) variables, where it consistently outperforms all baselines across all lead times. Compared to the current state-of-the-art model, ClimODE, PA–TFNP achieves lower RMSE, especially at longer horizons (18–24h), demonstrating improved temporal robustness.

Table 1: Comparison of RMSE values for baseline models and the proposed PA-TFNP (Ours) across different regions. The results highlight the performance differences for key atmospheric variables. Results are reported as mean ± standard deviation.

| Value | Hour | Australia | | | | South America | | | |
|---|---|---|---|---|---|---|---|---|---|
| | | NODE | ClimaX | ClimODE | **PA–TFNP** | NODE | ClimaX | ClimODE | **PA–TFNP** |
| z | 6 | 251.4 | 190.2 | 103.8 ± 14.6 | **79.5** ± 19.9 | 225.6 | 205.4 | 107.7 ± 20.2 | **87.5** ± 22.0 |
| | 12 | 344.8 | 184.7 | 170.7 ± 21.0 | **118.8** ± 30.1 | 365.6 | 220.2 | 169.4 ± 29.6 | **128.2** ± 31.3 |
| | 18 | 539.9 | 222.2 | 211.1 ± 31.6 | **161.6** ± 43.8 | 551.9 | 269.1 | 237.8 ± 32.2 | **174.1** ± 43.4 |
| | 24 | 632.7 | 324.9 | 308.2 ± 30.6 | **205.8** ± 59.5 | 660.3 | 301.8 | 292.0 ± 38.9 | **221.3** ± 57.8 |
| t | 6 | 1.37 | 1.19 | 1.05 ± 0.12 | **0.87** ± 0.14 | 1.58 | 1.38 | **0.97** ± 0.13 | 1.01 ± 0.16 |
| | 12 | 2.18 | 1.30 | 1.20 ± 0.16 | **1.07** ± 0.18 | 2.18 | 1.62 | 1.25 ± 0.18 | **1.18** ± 0.18 |
| | 18 | 2.68 | 1.39 | 1.33 ± 0.21 | **1.19** ± 0.20 | 2.74 | 1.79 | 1.43 ± 0.20 | **1.29** ± 0.18 |
| | 24 | 3.32 | 1.92 | 1.63 ± 0.24 | **1.31** ± 0.23 | 3.41 | 1.97 | 1.65 ± 0.26 | **1.44** ± 0.21 |
| t2m | 6 | 1.88 | 1.57 | **0.80** ± 0.13 | 2.42 ± 0.70 | 2.12 | 1.85 | **1.33** ± 0.26 | 1.73 ± 0.67 |
| | 12 | 2.02 | 1.57 | **1.10** ± 0.22 | 2.98 ± 1.50 | 2.42 | 2.08 | **1.04** ± 0.17 | 2.37 ± 1.20 |
| | 18 | 3.51 | 1.72 | **1.23** ± 0.24 | 2.37 ± 0.55 | 2.60 | 2.15 | **0.98** ± 0.17 | 1.87 ± 0.84 |
| | 24 | 2.46 | 2.15 | 1.25 ± 0.25 | **1.16** ± 0.24 | 2.56 | 2.23 | 1.17 ± 0.26 | **1.15** ± 0.27 |
| u10 | 6 | 1.91 | 1.40 | **1.35** ± 0.17 | 1.43 ± 0.19 | 1.94 | 1.27 | **1.25** ± 0.18 | 1.42 ± 0.27 |
| | 12 | 2.86 | 1.77 | 1.78 ± 0.21 | **1.74** ± 0.22 | 2.74 | 1.57 | **1.49** ± 0.23 | 1.56 ± 0.30 |
| | 18 | 3.44 | 2.03 | 1.96 ± 0.25 | **1.88** ± 0.26 | 3.24 | 1.83 | 1.81 ± 0.29 | **1.69** ± 0.29 |
| | 24 | 3.91 | 2.64 | 2.33 ± 0.33 | **2.06** ± 0.28 | 3.77 | 2.04 | 2.08 ± 0.35 | **1.86** ± 0.32 |
| v10 | 6 | 2.38 | 1.47 | **1.44** ± 0.20 | 1.56 ± 0.19 | 2.29 | 1.31 | **1.30** ± 0.21 | 1.68 ± 0.39 |
| | 12 | 3.60 | 1.79 | 1.87 ± 0.26 | **1.78** ± 0.25 | 3.42 | **1.64** | 1.71 ± 0.28 | 1.93 ± 0.40 |
| | 18 | 4.31 | 2.33 | 2.23 ± 0.23 | **2.04** ± 0.26 | 4.16 | 1.90 | 2.07 ± 0.31 | **1.88** ± 0.37 |
| | 24 | 4.88 | 2.58 | 2.53 ± 0.32 | **2.23** ± 0.30 | 4.76 | 2.14 | 2.43 ± 0.34 | **2.06** ± 0.37 |

For wind components, PA–TFNP slightly outperforms ClimODE in most settings, particularly at longer lead times. Notably, for t2m, PA–TFNP underperforms at earlier lead times but catches up or surpasses baselines at 24h. This may indicate a trade-off between local variance sensitivity and longer-horizon stability.

### 4.3 Monthly averaged weather forecasting

Next, we evaluate the predictive accuracy of ClimODE, CilmaX, TFNP, and PA-TFNP over a two-month lead time. All models predict the global two-month averaged future states based on an initial monthly average state. Table 2 provides a detailed comparison of RMSE values for various atmospheric variables, showing that PA-TFNP consistently outperforms other benchmarks, particularly in predicting geopotential height (z), atmospheric temperature (t) and ground temperature (t2m). The lower RMSE values in the results indicate that PA-TFNP more accurately captures complex climate patterns, offering enhanced reliability for extended-range climate forecasting.

### 4.4 Ablation Studies

**Assessing rotational equivariance: ClimODE vs TFNP.** To further evaluate the spatial prediction capabilities of TFNP, we compare its performance with ClimODE in terms of absolute prediction error across five key atmospheric variables (see Figure 6 in Appendix A). The results demonstrate that TFNP consistently achieves lower error magnitudes than ClimODE, particularly in geophysically challenging regions such as the poles and the equator. These regions are often prone to distortions due to their rotational properties, where ClimODE exhibits noticeable artifacts. In contrast, TFNP maintains strong spatial consistency, owing to its rotation-equivariant architecture. These findings underscore the importance of incorporating geometric inductive biases, such as rotational equivariance, in improving model robustness and accuracy in global-scale geophysical forecasting.

Table 2: Comparison of RMSE values for different models across two months. The results highlight the performance of TFNP and PA-TFNP compared to ClimODE and other baseline models for key atmospheric variables.

| Value | Months | ClimaX | ClimODE | TFNP (ours) | PA-TFNP (ours) |
|-------|--------|--------|---------|-------------|----------------|
| z | 1 | 580.73 | $692.10 \pm 119.80$ | $529.44 \pm 95.77$ | $\mathbf{502.01} \pm 79.50$ |
|   | 2 | 773.40 | $870.57 \pm 72.58$ | $527.07 \pm 84.54$ | $\mathbf{562.39} \pm 70.13$ |
| t | 1 | 2.89 | $2.81 \pm 0.48$ | $2.58 \pm 0.56$ | $\mathbf{2.48} \pm 0.45$ |
|   | 2 | 4.39 | $3.20 \pm 1.02$ | $\mathbf{2.42} \pm 0.42$ | $2.44 \pm 0.21$ |
| t2m | 1 | 2.97 | $4.33 \pm 0.38$ | $2.63 \pm 0.52$ | $\mathbf{2.53} \pm 0.34$ |
|   | 2 | 5.07 | $4.99 \pm 0.48$ | $2.95 \pm 0.45$ | $\mathbf{2.95} \pm 0.30$ |
| u10 | 1 | $\mathbf{1.80}$ | $1.98 \pm 0.19$ | $1.86 \pm 0.23$ | $1.83 \pm 0.23$ |
|   | 2 | $\mathbf{1.92}$ | $2.09 \pm 0.11$ | $2.40 \pm 0.22$ | $2.32 \pm 0.21$ |
| v10 | 1 | 1.50 | $1.66 \pm 0.18$ | $1.40 \pm 0.10$ | $\mathbf{1.39} \pm 0.12$ |
|   | 2 | $\mathbf{1.71}$ | $1.98 \pm 0.11$ | $1.95 \pm 0.18$ | $1.91 \pm 0.21$ |

**Benefits of Physics-Aware Modeling for Long-Term Stability: TFNP vs PA-TFNP.** To evaluate the effectiveness of Physics-Aware modeling, we compared the performance of the PA-TFNP model, which incorporate physical operators and features against the TFNP model. Experimental results shows that PA-TFNP consistently outperforms the TFNP model at extended forecast horizons beyond 24 hours, across all scalar quantities. These results underscore the importance of embedding physical properties within predictive models to achieve stable and reliable long-term forecasting, as clearly illustrated in Figure 4.

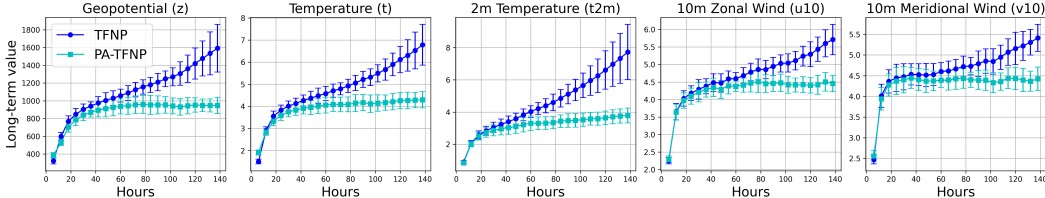

Figure 4: RMSE comparison of the TFNP baseline and Physics-Aware TFNP (PA-TFNP) models over extended forecast horizons (up to 138 hours) across multiple atmospheric variables (z, t, t2m, u10, v10). The PA-TFNP model, incorporating physical constraints, consistently demonstrates improved accuracy, highlighting the importance of physics-informed modeling for stable long-term predictions.

## 5 CONCLUSION AND LIMITATIONS

In this work, we have presented the Physics-Aware TFNP, a novel framework that combines deep learning with fundamental physical principles to tackle climate and weather prediction tasks more accurately and robustly. By integrating gradient computation and boundary treatment methods rooted in numerical techniques and by incorporating physically consistent diffusion terms and divergence-free conditions, our approach addresses the shortcomings of both purely data-driven and physics-agnostic models. TFNP not only demonstrates state-of-the-art forecasting performance but also maintains physical fidelity, offering enhanced interpretability and reliability. We anticipate that the mathematical principles introduced here will generalize across a broad range of scientific computing domains, thereby accelerating progress in both global and regional weather prediction.

As expected, the rotation-equivariant feature of the proposed PA-TFNP plays an important role in the global forecasting model. However, this characteristic appears to offer limited benefits for regional forecasting. This limitation warrants further investigation in future work. We have added diffusion terms to the model equations for all predictive variables. However, the modification of the model equation should be tailored to each variable, as their physical interpretations differ significantly. For instance, the temperature variable and ground wind variables represent fundamentally different physical phenomena and therefore should be modeled using distinct equations.

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

## A  FURTHER EXPERIMENTS

In this section, we present additional experimental results for the TFNP and the PA-TFNP model. Table 2 and Figure 5 report the RMSE values for the two-month prediction task described in Section 4.3. Figure 5 visualizes the RMSE values of ClimODE, TFNP, and PA-TFNP over a two-month forecast horizon, based on global monthly averaged predictions. The comparison spans five atmospheric variables ($z$, $t$, $t2m$, $u10$, and $v10$). The results clearly show that PA-TFNP achieves the lowest RMSE in most variables, especially in $z$ and $t$, where its advantage over other models is more pronounced. This supports our main claim that incorporating physics-aware inductive bias improves long-range prediction performance.

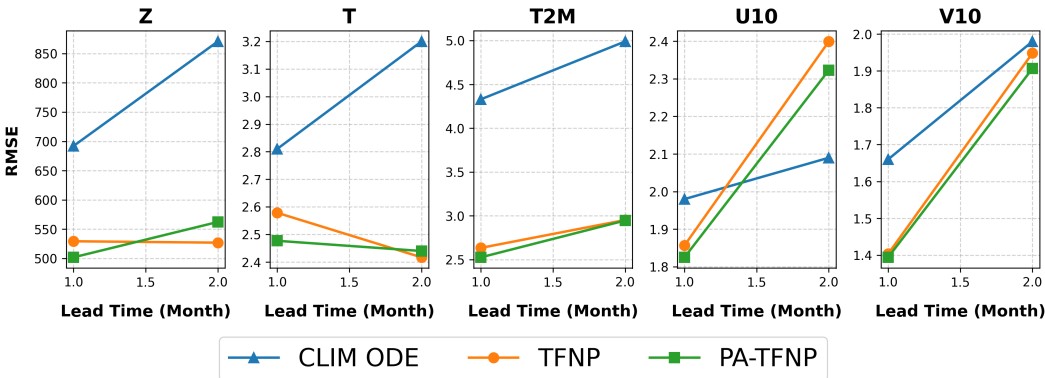

Figure 5: RMSE comparison of ClimODE, TFNP, and PA-TFNP for two-month averaged predictions across five key atmospheric variables: geopotential height ($z$), temperature ($t$), ground temperature ($t2m$), and wind components ($u10$, $v10$). PA-TFNP consistently achieves the lowest RMSE for most variables, particularly in $z$ and $t$, demonstrating enhanced accuracy and temporal stability for long-range climate forecasting.

Figure 6 shows the absolute prediction errors of PA-TFNP and ClimODE across five key atmospheric variables, as discussed in Section 4.3.

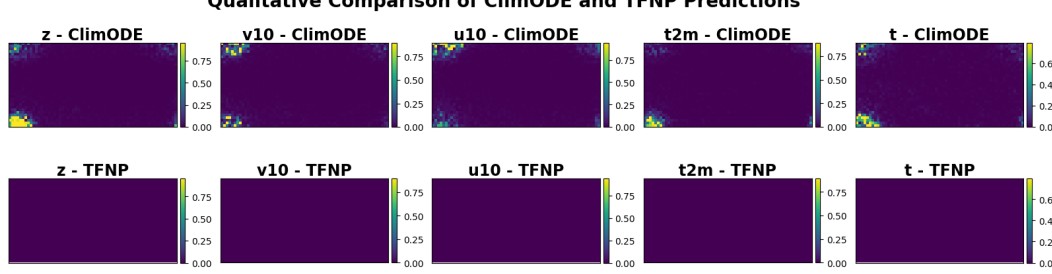

Figure 6: Qualitative comparison of absolute prediction errors from ClimODE and TFNP across five atmospheric variables ($z$, $v10$, $u10$, $t2m$, $t$). The first row visualizes the spatial distribution of prediction errors from ClimODE, while the second row shows those from TFNP. TFNP significantly reduces errors and improves spatial consistency, particularly in polar regions where ClimODE suffers from grid distortion effects. These results highlight TFNP's ability to handle rotationally sensitive areas through its rotation-equivariant architecture, enabling more robust global-scale predictions.

Next, in Table 3, we compare the PA-TFNP model with other baseline models—Neural ODE, ClimaX, and ClimODE—for the North America region. See Table 1 for results in other regions, including Australia and South America.

Figure 7 and Figure 8 present qualitative visualizations of PA-TFNP's prediction performance for monthly-averaged and hourly forecasting tasks, respectively. Across both long-term (monthly) and

Table 3: Comparison of RMSE values for baseline models and the proposed PA-TFNP (Ours) across North America. Results are reported as mean $\pm$ standard deviation.

| Value | Hours | NODE | ClimaX | ClimODE | PA–TFNP (Ours) |
|-------|-------|------|--------|---------|----------------|
| z | 6 | 232.8 | 273.4 | $134.5 \pm 10.6$ | $\mathbf{130.2} \pm 41.3$ |
| | 12 | 469.2 | 329.5 | $225.0 \pm 17.3$ | $\mathbf{202.3} \pm 59.1$ |
| | 18 | 667.2 | 543.0 | $307.7 \pm 25.4$ | $\mathbf{282.7} \pm 81.8$ |
| | 24 | 893.7 | 494.8 | $390.1 \pm 32.3$ | $\mathbf{367.3} \pm 107.6$ |
| t | 6 | 1.96 | 1.62 | $\mathbf{1.28} \pm 0.06$ | $1.45 \pm 0.27$ |
| | 12 | 3.34 | 1.86 | $1.81 \pm 0.13$ | $\mathbf{1.79} \pm 0.37$ |
| | 18 | 4.21 | 2.75 | $2.03 \pm 0.16$ | $\mathbf{1.97} \pm 0.43$ |
| | 24 | 5.39 | 2.27 | $\mathbf{2.23} \pm 0.18$ | $2.32 \pm 0.48$ |
| t2m | 6 | 2.65 | 1.75 | $\mathbf{1.61} \pm 0.12$ | $3.48 \pm 1.74$ |
| | 12 | 3.43 | 1.87 | $\mathbf{1.87} \pm 0.13$ | $4.68 \pm 1.03$ |
| | 18 | 3.53 | 2.27 | $\mathbf{1.96} \pm 0.33$ | $3.41 \pm 1.05$ |
| | 24 | 3.39 | 1.93 | $\mathbf{2.15} \pm 0.20$ | $2.59 \pm 0.64$ |
| u10 | 6 | 1.96 | 1.74 | $\mathbf{1.54} \pm 0.19$ | $1.69 \pm 0.34$ |
| | 12 | 2.91 | 2.24 | $2.01 \pm 0.20$ | $\mathbf{1.94} \pm 0.41$ |
| | 18 | 3.40 | 3.42 | $2.17 \pm 0.34$ | $\mathbf{2.08} \pm 0.43$ |
| | 24 | 3.96 | 3.42 | $2.34 \pm 0.32$ | $\mathbf{2.26} \pm 0.43$ |
| v10 | 6 | 2.36 | 1.83 | $\mathbf{1.67} \pm 0.23$ | $1.79 \pm 0.36$ |
| | 12 | 3.42 | 2.43 | $2.03 \pm 0.31$ | $\mathbf{1.94} \pm 0.41$ |
| | 18 | 4.35 | 3.92 | $2.31 \pm 0.37$ | $\mathbf{2.20} \pm 0.40$ |
| | 24 | 4.57 | 3.39 | $2.50 \pm 0.41$ | $\mathbf{2.37} \pm 0.42$ |

short-term (hourly) settings, PA-TFNP demonstrates low absolute errors across the entire spatial domain, including boundary regions. The model consistently provides accurate predictions for key atmospheric variables, particularly temperature ($t$) and geopotential height ($z$), underscoring its effectiveness in spatiotemporal climate modeling.

# B DATASETS

The ERA5 dataset consists of weather records for five variables: round temperature (t2m), atmospheric temperature (t), geopotential height (z), and ground wind components (u10, v10). It provides global coverage on a uniform grid from 2006 to 2018. Data from 2006 to 2015 are used for training, 2016 for validation, and 2017-2018 for testing. We exclude the first two and last one months of each year, considering only nine months per year. These months are further divided into three sequential groups, where, for each group, we predicted atmospheric variables for two consecutive months based on observations from the preceding month. The spatial grid is uniformly spaced at $5.625°$ in both latitude and longitude, with dimensions $H = 32$ and $W = 64$.

# C TRAINING DETAILS

We employed the forward Euler method as our ODE solver to integrate the dynamical system in Equation (1) and its variation, using a time resolution of 1/6 month (approximately 5 days). In our neural ODE framework, this resolution is represented as 0.01 in normalized time to avoid excessive computational costs of directly using a one-month unit. Model training and inference are performed on a single NVIDIA RTX 4090 (24GB). All training hyperparameters for ClimODE and ClimaX remain consistent with those in (Nguyen et al., 2023; Verma et al., 2024). During training, all variables are normalized to [0,1] using min-max scaling; however, the original values are restored to compute RMSD in Table 2.

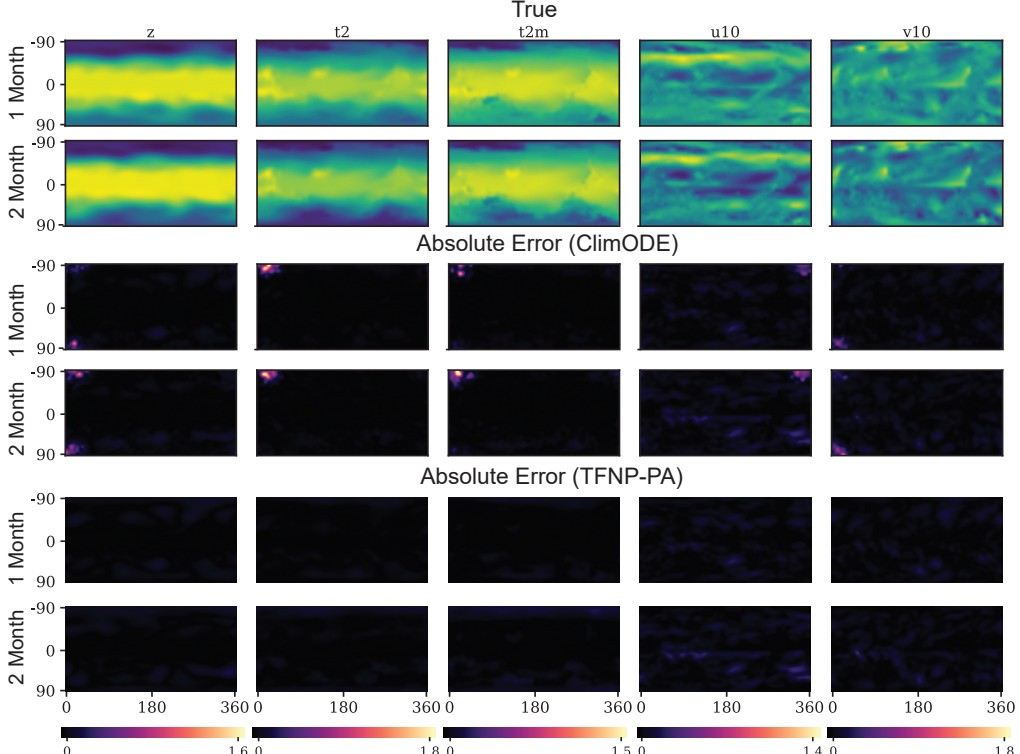

Figure 7: Comparison of RMSE values for ClimODE and TFNP-PA in two-month averaged predictions for five atmospheric variables. TFNP consistently achieves lower RMSE, particularly for temperature (t) and geopotential height (z), highlighting its improved forecasting performance.

## C.1 LOSS FUNCTION

Consider the set of observations $\{\{q_i^{obs}(\mathbf{x}_n, t)\}_{i=1}^d\}_{n=1}^N$. To introduce stochasticity, we assume the following equation with the estimated mean and variance as in (Verma et al., 2024).:

$$q_i^{\text{obs}}(\mathbf{x}, t) \sim \mathcal{N}\left(q_i(\mathbf{x}, t) + \mu_i(\mathbf{x}, t), \sigma_i^2(\mathbf{x}, t)\right),$$

where the function $\epsilon\left(q_i(\mathbf{x}, t), g(\{x_n\}_{n=1}^N, t)\right)$ estimates the additional mean $\mu_i(\mathbf{x}, t)$ and variance $\sigma_i^2(\mathbf{x}, t)$. Given the observations $\{q_i^{obs}(\mathbf{x}_n, t)\}$, we define the loss function $\mathcal{L}(\eta)$ using the negative log-likelihood:

$$\log L(\{\{q_i^{obs}(\mathbf{x}_n, t)\}_{i=1}^d\}_{n=1}^N | \{\{\mu_i(\mathbf{x}_n, t)\}_{i=1}^d\}_{n=1}^N, \{\{\sigma_i(\mathbf{x}_n, t)\}_{i=1}^d\}_{n=1}^N)$$

$$= \log \prod_{i=1}^d \prod_{n=1}^N L(q_i^{obs}(\mathbf{x}_n, t) | \mu_i(\mathbf{x}_n, t), \sigma_i(\mathbf{x}_n, t))$$

$$= \sum_{i=1}^d \sum_{n=1}^N \log(L(q_i^{obs}(\mathbf{x}_n, t) | \mu_i(\mathbf{x}_n, t), \sigma_i(\mathbf{x}_n, t)))$$

$$= \sum_{i=1}^d \sum_{n=1}^N \left[ \frac{(q_i^{obs}(\mathbf{x}_n, t) - \mu_i(\mathbf{x}_n, t))^2}{2(\sigma_i(\mathbf{x}_n, t))^2} + \log\left(\sqrt{2\pi}\sigma_i(\mathbf{x}_n, t)\right) \right].$$

Table 4: Comparison of RMSE values for ClimODE and the proposed PA-TFNP (Ours) across two spatiotemporal resolutions. The results highlight the performance differences for key atmospheric variables. Results are reported as mean $\pm$ standard deviation.

| (a) Long-term prediction at a resolution of $5.625°$. | | | | (b) Short-term prediction at a resolution of $11.25°$. | | |
|---|---|---|---|---|---|---|
| Value | Day | ClimODE | TFNP (Ours) | Hour | ClimODE | TFNP (Ours) |
| | 5 | $1104.0 \pm 104.0$ | $\mathbf{220.4} \pm 21.6$ | 6 | $3115.1 \pm 216.6$ | $\mathbf{161.2} \pm 17.4$ |
| | 10 | $1445.7 \pm 103.1$ | $\mathbf{988.3} \pm 91.1$ | 12 | $3156.4 \pm 204.0$ | $\mathbf{602.5} \pm 59.6$ |
| | 15 | $1430.2 \pm 118.6$ | $\mathbf{1033.3} \pm 87.5$ | 18 | $3175.8 \pm 176.3$ | $\mathbf{830.3} \pm 77.1$ |
| z | 20 | $1470.3 \pm 129.0$ | $\mathbf{1058.2} \pm 70.9$ | 24 | $3202.5 \pm 176.5$ | $\mathbf{935.5} \pm 87.8$ |
| | 25 | $1445.6 \pm 83.6$ | $\mathbf{1018.0} \pm 81.1$ | 30 | $3240.1 \pm 178.9$ | $\mathbf{999.8} \pm 90.2$ |
| | 30 | $1449.0 \pm 108.5$ | $\mathbf{1014.4} \pm 92.8$ | 36 | $3216.3 \pm 181.3$ | $\mathbf{1038.1} \pm 90.7$ |
| | 35 | $1457.8 \pm 113.1$ | $\mathbf{1078.0} \pm 68.6$ | 42 | $3282.4 \pm 224.7$ | $\mathbf{1067.6} \pm 106.3$ |
| | 5 | $6.12 \pm 0.55$ | $\mathbf{1.15} \pm 0.13$ | 6 | $22.62 \pm 1.41$ | $\mathbf{1.27} \pm 0.28$ |
| | 10 | $7.58 \pm 0.45$ | $\mathbf{4.34} \pm 0.47$ | 12 | $22.73 \pm 1.16$ | $\mathbf{3.22} \pm 0.21$ |
| | 15 | $7.73 \pm 0.86$ | $\mathbf{4.49} \pm 0.33$ | 18 | $22.59 \pm 1.07$ | $\mathbf{3.99} \pm 0.28$ |
| t | 20 | $7.46 \pm 0.56$ | $\mathbf{4.54} \pm 0.30$ | 24 | $22.49 \pm 1.17$ | $\mathbf{4.29} \pm 0.30$ |
| | 25 | $7.67 \pm 0.58$ | $\mathbf{4.64} \pm 0.46$ | 30 | $22.50 \pm 1.09$ | $\mathbf{4.49} \pm 0.31$ |
| | 30 | $8.07 \pm 0.83$ | $\mathbf{4.70} \pm 0.42$ | 36 | $22.51 \pm 1.13$ | $\mathbf{4.61} \pm 0.35$ |
| | 35 | $8.16 \pm 0.51$ | $\mathbf{4.98} \pm 0.42$ | 42 | $22.86 \pm 1.15$ | $\mathbf{4.71} \pm 0.37$ |
| | 5 | $7.92 \pm 0.71$ | $\mathbf{1.16} \pm 0.13$ | 6 | $38.58 \pm 4.77$ | $\mathbf{0.92} \pm 0.17$ |
| | 10 | $8.78 \pm 0.82$ | $\mathbf{3.09} \pm 0.37$ | 12 | $39.03 \pm 4.70$ | $\mathbf{2.27} \pm 0.18$ |
| | 15 | $8.58 \pm 0.80$ | $\mathbf{3.43} \pm 0.38$ | 18 | $38.48 \pm 4.12$ | $\mathbf{2.80} \pm 0.26$ |
| t2m | 20 | $8.76 \pm 0.87$ | $\mathbf{3.59} \pm 0.33$ | 24 | $38.19 \pm 4.50$ | $\mathbf{3.04} \pm 0.28$ |
| | 25 | $8.90 \pm 0.95$ | $\mathbf{3.81} \pm 0.45$ | 30 | $39.19 \pm 3.85$ | $\mathbf{3.20} \pm 0.31$ |
| | 30 | $9.09 \pm 0.70$ | $\mathbf{4.04} \pm 0.48$ | 36 | $39.21 \pm 4.21$ | $\mathbf{3.33} \pm 0.34$ |
| | 45 | $9.51 \pm 1.23$ | $\mathbf{4.30} \pm 0.62$ | 42 | $37.87 \pm 4.35$ | $\mathbf{3.42} \pm 0.35$ |
| | 5 | $3.99 \pm 0.30$ | $\mathbf{0.96} \pm 0.08$ | 6 | $14.86 \pm 0.92$ | $\mathbf{0.62} \pm 0.04$ |
| | 10 | $6.29 \pm 0.32$ | $\mathbf{4.86} \pm 0.25$ | 12 | $15.44 \pm 0.92$ | $\mathbf{3.98} \pm 0.26$ |
| | 15 | $6.27 \pm 0.29$ | $\mathbf{5.17} \pm 0.24$ | 18 | $15.57 \pm 0.92$ | $\mathbf{4.69} \pm 0.27$ |
| u10 | 20 | $6.09 \pm 0.20$ | $\mathbf{5.18} \pm 0.22$ | 24 | $15.67 \pm 0.83$ | $\mathbf{4.97} \pm 0.30$ |
| | 25 | $6.08 \pm 0.31$ | $\mathbf{5.12} \pm 0.21$ | 30 | $15.70 \pm 0.88$ | $\mathbf{5.12} \pm 0.35$ |
| | 30 | $6.09 \pm 0.30$ | $\mathbf{5.11} \pm 0.26$ | 36 | $15.86 \pm 0.82$ | $\mathbf{5.23} \pm 0.35$ |
| | 35 | $6.22 \pm 0.27$ | $\mathbf{5.21} \pm 0.21$ | 42 | $15.89 \pm 0.87$ | $\mathbf{5.32} \pm 0.35$ |
| | 5 | $3.94 \pm 0.44$ | $\mathbf{1.50} \pm 0.08$ | 6 | $13.86 \pm 0.92$ | $\mathbf{0.59} \pm 0.05$ |
| | 10 | $6.41 \pm 0.29$ | $\mathbf{4.83} \pm 0.28$ | 12 | $14.46 \pm 0.92$ | $\mathbf{4.50} \pm 0.36$ |
| | 15 | $6.19 \pm 0.37$ | $\mathbf{4.98} \pm 0.38$ | 18 | $14.74 \pm 0.82$ | $\mathbf{5.17} \pm 0.37$ |
| v10 | 20 | $6.18 \pm 0.37$ | $\mathbf{5.00} \pm 0.23$ | 24 | $14.75 \pm 0.74$ | $\mathbf{5.32} \pm 0.37$ |
| | 25 | $5.93 \pm 0.29$ | $\mathbf{4.94} \pm 0.30$ | 30 | $14.78 \pm 0.84$ | $\mathbf{5.36} \pm 0.36$ |
| | 30 | $6.29 \pm 0.39$ | $\mathbf{4.88} \pm 0.28$ | 36 | $14.73 \pm 0.74$ | $\mathbf{5.39} \pm 0.33$ |
| | 35 | $6.21 \pm 0.31$ | $\mathbf{4.97} \pm 0.28$ | 42 | $14.69 \pm 0.92$ | $\mathbf{5.43} \pm 0.36$ |

To enhance numerical stability, we incorporate a small constant $10^{-3}$ into the variance term and introduce a regularization term weighted by $\lambda$:

$$\sum_{i=1}^{d} \sum_{n=1}^{N} \left[ \frac{(q_i^{obs}(\mathbf{x}_n, t) - \mu_i(\mathbf{x}_n, t))^2}{2(\sigma_i(\mathbf{x}_n, t))^2 + 10^{-3}} + \log(\sigma_i(\mathbf{x}_n, t) + 10^{-3}) \right] + \lambda \sum_{i=1}^{d} \sum_{n=1}^{N} (\sigma_i^2(\mathbf{x}_n, t))$$

## C.2 LATITUDE-WEIGHTED RMSE METRIC

To quantify prediction accuracy, we employ the latitude-weighted RMSE metric, defined as:

$$\text{RMSE} = \frac{1}{T} \sum_{t}^{T} \sqrt{\frac{1}{HW} \sum_{h}^{H} \sum_{w}^{W} \alpha(h)(y_{thw} - u_{thw})^2}$$

where $\alpha(h) = \cos(h) / \frac{1}{H} \sum_{h'}^{H} \cos(h')$ represents the latitude-dependent weighting factor.

Table 5: Training time for 1 epoch and the number of parameters.

| Category | Model | Time [s] | #Params |
|---|---|---|---|
| **All-data** | ClimaX | | 115M |
| **Regional** | | | |
| North | ClimODE | 305.00 | 2.75M |
| | TFNP (Ours) | 289.87 | 2.78M |
| South | ClimODE | 309.86 | 2.75M |
| | TFNP (Ours) | 295.46 | 2.78M |
| Australia | ClimODE | 310.08 | 2.75M |
| | TFNP (Ours) | 292.43 | 2.78M |
| **Global** | ClimODE | 23.69 / 55.60 | 2.75M |
| Long Term / High Resolution | PA-TFNP (Ours) | 11.27 / 31.39 | 0.196M |
| **Monthly** | ClimODE | 6.50 | 2.40M |
| | TFNP (Ours) | 2.87 | 0.098M |
| | PA-TFNP (Ours) | 3.30 | 0.194M |
| **Ablation** | TFNP (Ours) | 3.17 | 0.130M |
| | PA-TFNP (Ours) | 4.38 | 0.196M |

## C.3 COMPUTATIONAL COST

Table 5 reports the training time per epoch and the number of trainable parameters for various models and experimental settings. Our proposed models (TFNP and PA-TFNP) demonstrate notable efficiency in both training speed and model size compared to baseline models like ClimODE and ClimaX. In regional settings, our models consistently show faster training times despite having a comparable number of parameters. For global high-resolution forecasts, PA-TFNP achieves significantly reduced training time, with the number of parameters less than 10% of ClimODE's, highlighting its scalability. Furthermore, under the monthly and ablation settings, our lightweight PA-TFNP remains both computationally efficient and parameter-efficient, making it suitable for practical deployment in climate modeling tasks.

## D ADDITIONAL EXPLANATION ON PHYSICS-AWARE VARIANTS

This section details the physics-informed variant model presented in Section 3.3.

### RATIONALE FOR ADDING PHYSICAL TERMS

Atmospheric phenomena, such as turbulence, can distribute energy across scales—an effect we attempt to approximate with the Laplacian $\Delta$ term. Even when the governing equations do not explicitly include diffusion, some level of numerical diffusion is generally needed to prevent the accumulation of artificial energy in simulations; see (Warner, 2010, Section 3.4.7). To address these practical considerations, we include a term $\alpha\Delta(\cdot)$ in the original transport equation equation 1, where the non-negative coefficient $\alpha$ is a learnable parameter. Setting $\alpha = 0$ preserves the original equation equation 1, while a small $\alpha > 0$ allows the model to represent physically motivated diffusion in a controlled manner.

Similarly, the physics-informed forcing term $f_{\text{phys}}(\mathbf{x}, t, \mathbf{u}i)$ explicitly aims to incorporate essential physical processes into the neural equation. We define

$$f\text{phys}(\mathbf{x}, t, \mathbf{u}_i) = -\nabla\Phi + \nu\Delta\mathbf{u}_i - \gamma\mathbf{u}_i,$$

where $\Phi$ represents the geopotential field (with $\Phi = z$), and $\nu$, $\gamma$ are learnable coefficients associated with viscosity and linear drag, respectively. Including these physical terms is intended to enhance the consistency of the model with fundamental atmospheric physics. Our experiment shows that our approach improves interpretability, prediction accuracy, and numerical stability, particularly in longer-range atmospheric simulations.

Thus, our hybrid formulation of PA-TFNP aims to retain the expressive capability of neural network models while encouraging adherence to important physical constraints, potentially leading to improved predictive performance for long-term forecasts.

### MOTIVATION OF ADDITIONAL PHYSICS-DERIVED FEATURES

We introduce three physics-derived features and recall the governing relations in which each enters:

- **Wind magnitude.** $|V_{10}| = \sqrt{u_{10}^2 + v_{10}^2}$ appears in the bulk aerodynamic surface stress formula
$$\tau = \rho\, C_D\, |V_{10}|\, V_{10},$$
where $\tau$ is asurface stress vector, $\rho$ is air density and $C_D$ a drag coefficient.

- **Lapse rate.** $\Delta t = t - t_{2m}$ appears in various governing equations, particularly in parameterizations of turbulent mixing processes in the atmospheric boundary layer. Specifically, it is utilized to estimate buoyancy-driven turbulence production or suppression, represented mathematically by terms such as:
$$B \propto -\frac{g}{\theta_0}\, K_H\, \Delta t,$$
where $B$ denotes the buoyancy contribution to the turbulent kinetic energy budget, $g$ is gravitational acceleration, $\theta_0$ represents a reference potential temperature, and $K_H$ is the eddy diffusivity for heat. Therefore, the lapse rate $\Delta t$ can be considered a valuable physical feature when developing approximate neural network-based flow models, as it directly encapsulates critical information about atmospheric stability and turbulence dynamics.

- **Relative vorticity.** $\zeta = \partial_y v_{10} - \partial_x u_{10}$ quantifies the local horizontal spin of the wind and features explicitly in the barotropic vorticity equation, quasi-geostrophic potential vorticity, and Ertel potential vorticity. Its direct link to these conservation laws makes $\zeta$ a clear, physically interpretable variable for weather-prediction modeling.

### SPATIAL DERIVATIVE

The spatial derivative approximation used in this study is based on the gradient operator in spherical coordinates. Let $F(R, \phi, \lambda)$ be a scalar field defined on the surface of a sphere, where $\phi$ denotes latitude (in radians), $\lambda$ denotes longitude (in radians), and $R$ represents the Earth's radius, which is assumed to be constant throughout the domain. Although the notation $\mathbb{S}^2$ typically refers to the unit sphere, here we consider the spherical surface of radius $R$, denoted $\mathbb{S}_R^2 = \{\mathbf{x} \in \mathbb{R}^3 : \|\mathbf{x}\| = R\}$. Assuming no variation in the radial direction, the surface gradient takes the following form:

$$\nabla_{\mathbb{S}_R^2} F(\phi, \lambda) = \frac{1}{R} \frac{\partial F}{\partial \phi} \hat{e}_\phi + \frac{1}{R\cos\phi} \frac{\partial F}{\partial \lambda} \hat{e}_\lambda.$$

Here, $\hat{e}_\phi$ is the unit vector in the direction of increasing latitude (northward), and $\hat{e}_\lambda$ is the unit vector in the direction of increasing longitude (eastward). Both lie in the tangent plane of the sphere at each point.

Based on this formulation, the gradient is numerically approximated using a second-order central finite difference scheme, yielding Equation equation 3, where the factor $\pi/180$ converts angular increments from degrees to radians. The spherical derivative approximation is applied consistently throughout the PA-TFNP model.

## E BROADER IMPACT

The proposed PA-TFNP improves global and regional weather prediction by combining physical interpretability with high forecasting accuracy and reduced computational cost. This enables faster, more accessible predictions, especially valuable in regions with limited computing resources. The model's efficiency supports broader deployment of weather forecasting systems, contributing to better preparedness for climate change. Careful integration with traditional methods and responsible communication are essential for safe and effective use.

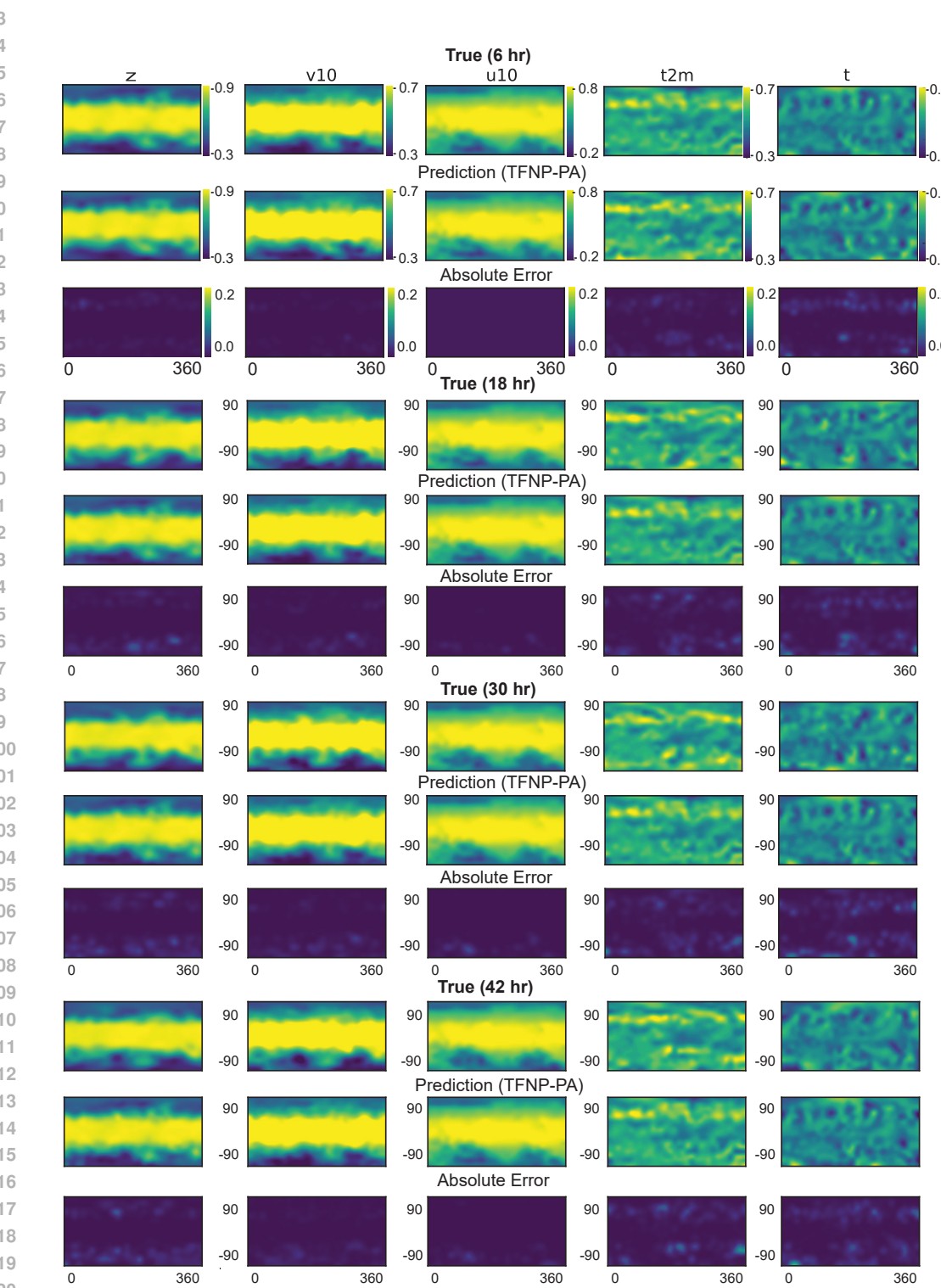

Figure 8: Comparison of RMSE values for TFNP-PA in long-term predictions over hourly predictions for five atmospheric variables. TFNP consistently predicts over the entire domain, particularly for temperature (t) and geopotential height (z), highlighting its improved forecasting performance.

