# OpenReview forum: "Physics-Aware Tensor Field Neural PDE for Climate and Weather Prediction"
_ICLR.cc/2026/Conference — ICLR 2026 Conference Withdrawn Submission_

### Official Review · Reviewer_bGma · 2025-10-20

**Soundness:** 3
**Presentation:** 3
**Contribution:** 2
**Rating:** 2
**Confidence:** 3

**Summary:**

The paper introduces a physics-informed deep learning framework for climate and weather prediction. Unlike purely data-driven models (e.g., ClimODE, ClimaX), PA-TFNP embeds rotation-equivariant tensor-field neural operators directly on the sphere (to respect Earth’s geometry), incorporates spherical-transform-based gradient operators and physically consistent boundary treatments, and adds diffusion and momentum correction terms derived from the atmospheric primitive equations for long-term stability. Across global, regional, and monthly prediction tasks on ERA5 data, PA-TFNP achieves up to 78.9% RMSE reduction over ClimODE with 10× fewer parameters.

**Strengths:**

1. The idea is reasonable. Combining rotation-equivariant tensor fields, spherical gradients, and physics-based diffusion is a good blend of geometry, physics, and learning. The novelty is very incremental, concatenating ideas from many previous works, but reasonable.

2. Demonstrates consistent and substantial accuracy gains across multiple spatiotemporal scales (hourly → monthly, regional → global). Clear modular structure (TFNP → PA-TFNP), with derivations and ablations validating each design choice.

3. Addresses the pressing need for physically consistent ML weather models, aligning well with current research trends (e.g., GraphCast, Aurora).

**Weaknesses:**

1. No code revealed for reproducibility check. The implementation details and reproducibility remain limited. Important training hyperparameters, computational costs at large scale, and code availability are not discussed in depth. These omissions make it challenging for others to replicate or extend the proposed approach.

2. While the model introduces diffusion and drag terms inspired by the atmospheric primitive equations, these additions are applied uniformly across all variables. This uniform treatment overlooks the fundamental physical differences between quantities such as temperature, wind, and geopotential height. Each obeys distinct physical laws and should ideally have customized formulations rather than a single shared diffusion process.

3. The benefits of rotation-equivariance, though central to the model’s innovation, appear to be confined largely to global-scale forecasting, and have been used by many previous works already. The only physics-aware part is according to the advection structure design of the PDE, which is also not new. ClimODE did the same already. **This is the main reason for me recommending paper reject**. The authors should try rebuttal on this first before everything else.

4. While comparisons to ClimODE and ClimaX are thorough, the paper omits quantitative benchmarks against more recent large-scale foundation models like Aurora or GraphCast, both of which represent strong baselines in this field. Including such comparisons would strengthen the empirical claims of superiority. ClimODE is well-known to not perform well enough compared to large FMs, and its core contribution is more on guiding people to use physics and AI inherently together by methods other than just tuning loss functions. Simply showing an advantage over ClimODE is not entirely surprising or impactful, unless novel techniques/paradigms are proposed.

5. Another limitation lies in the evaluation metrics. The study relies primarily on RMSE, without assessing physically meaningful diagnostics such as conservation of mass, energy, or vorticity, nor does it provide uncertainty estimates. This omission makes it difficult to judge whether the model’s forecasts are physically consistent, beyond being numerically accurate.

**Questions:**

1. How is the spatially varying diffusion coefficient α(x) learned? Is it through per grid point or a continuous modeling?

2. How sensitive is performance to the blending schedule τ₀? Could it be dynamically adapted instead of fixed exponential decay?

3. How does PA-TFNP perform on 0.25° or sub-hourly resolutions? Does rotation-equivariance introduce computational overhead?

4. Have you evaluated transfer performance to other reanalysis datasets or regional NWP outputs?

5. Can tensor features or learned diffusion fields be visualized to reveal interpretable atmospheric patterns (e.g., jet streams, vortices)? Can you show some examples?

---

### Official Review · Reviewer_h8TH · 2025-11-01

**Soundness:** 2
**Presentation:** 2
**Contribution:** 1
**Rating:** 2
**Confidence:** 4

**Summary:**

The paper proposes Physics-Aware Tensor Field Neural PDE (PA-TFNP), a new neural ODE framework for weather forecasting. PA-TFNP embeds rotation-equivariant tensor-field neural operators on the sphere, employs a gradient operator based on spherical transforms, and augments the learned dynamics with diffusion terms derived from the atmospheric primitive equations. The proposed method outperforms ClimODE and ClimaX on global and regional weather forecasting tasks.

**Strengths:**

- The paper is original to the best of my knowledge.

**Weaknesses:**

My main concern is regarding the significance of the work.
- The experiment setting is far from standard: ClimaX and ClimODE are not representative baselines since they are inferior to IFS-HRES and many recent methods: Graphcast, Pangu-Weather, FourcastNet3, Stormer, etc. Moreover, using Weatherbench is no longer the standard, since the updated version of the benchmark was already published two years ago. The authors used the lowest resolution data in WB (5.625°), and even its lower resolution version (11.25°), which is not up to the standard of other works in this field.
- The main experimental result in Figure 3 is also poorly presented: apart from lacking strong baselines, it also does not show Climatology and IFS-HRES, two must-have baselines in weather forecasting. Moreover, I am confused why the authors evaluated short-term forecasts for the 11.25° data and long-term forecasts for the 5.625° data instead of plotting for all lead times, and why do we need evaluation at an additional lower resolution of 11.25°?
- The empirical performance of the proposed method does not make much sense. Both ClimODE and the proposed method perform extremely poorly across various variables, being worse than Climatology at day 10 for the 5.625° data, and even at 12 hours lead time for the 11.25° data. This is nowhere near the performance of modern methods.
- The introduction brings up generalization across diverse geophysical scenarios and error accumulation as the weaknesses of existing methods, but does not show in the paper how the proposed method resolves these problems.

Other comments:
- The paper does not cite many recent works in deep learning for weather forecasting: Fuxi [1], NeuralGCM [2], Stormer [3], Gencast [4], and Omnicast [5], just to name a few. All of these methods are competitive with IFS-HRES and should be the standard baselines to compare with.

[1] Chen, Lei, et al. "FuXi: a cascade machine learning forecasting system for 15-day global weather forecast." npj climate and atmospheric science 6.1 (2023): 190.

[2] Kochkov, Dmitrii, et al. "Neural general circulation models for weather and climate." Nature 632.8027 (2024): 1060-1066.

[3] Nguyen, Tung, et al. "Scaling transformer neural networks for skillful and reliable medium-range weather forecasting." Advances in Neural Information Processing Systems 37 (2024): 68740-68771.

[4] Price, Ilan, et al. "Probabilistic weather forecasting with machine learning." Nature 637.8044 (2025): 84-90.

[5] Nguyen, Tung, et al. "OmniCast: A Masked Latent Diffusion Model for Weather Forecasting Across Time Scales." arXiv preprint arXiv:2510.18707 (2025).

**Questions:**

- What are the benefits of the proposed method compared to existing methods?

---

### Official Review · Reviewer_rRuZ · 2025-11-01

**Soundness:** 3
**Presentation:** 3
**Contribution:** 2
**Rating:** 2
**Confidence:** 3

**Summary:**

This paper proposes the Physics-Aware Tensor Field Neural PDE (PA-TFNP), a forecasting framework that embeds rotation-equivariant tensor-field neural operators directly on the sphere with well-designed physical constraints being incorporated. Experiments on 5.625 degree and 11.25 degree shows improved performance in comparison with ClimODE and ClimX.

Overall, this paper can be considered an incremental work. Indeed, it addresses the challenges in the poles and the equator. However, how the performance improvement led by the proposed mechanism remains unclear to me. I like the idea proposed in this paper, but the weak and incomplete experiments make the empirical justification weak.

**Strengths:**

1.	To solve the geometric distortions near the polar regions, this paper proposes a Tensor Field Neural PDE framework (TFNP) powered by tensor-field neural networks. It can capture rotationally equivariant spatiotemporal patterns and leads to improved weather prediction.

2.	It further incorporates physical constraints into TFNP, leading to the so-called PA-TFNP, including 1) applying boundary conditions reflecting physical properties; 2) computing spatial derivatives using spherical operators; 3) augmenting inputs to $f_\eta$ with physical features.

**Weaknesses:**

1.	This paper can be considered as an incremental work of existing work. How this method advances the AI-based weather forecasting in the literature is not very clear to me, especially considering its weak performance in experiments.

2.	Limited experiments. The settings of the experiments are flawed. Only 5.625 degree and 11.25 degree. The comparison in Table 2 only presents 5 meteorological variables. The baseline algorithms considered are ClimX and ClimODE, which are not SOTA algorithms. I woud recommend the authors follow the settings of WeatherBench2 and compare the RMSE with other SOTA algorithms (please refer to https://sites.research.google/gr/weatherbench/deterministic-scores/). In Figure 3, the baseline of ClimODE is too weak. The RMSE of t2m is above 8.0 (check the top middle plot of Figure 3), which is much worse than the climatology.  For Table 1, the RMSEs of t2m are all greater than 1.0 in the 24 hours for all compared algorithms. According to WeatherBench2, most SOTA algorithms for t2m (please go to https://sites.research.google/gr/weatherbench/deterministic-scores/, and select 2m Temperature variable, and select resolution 64 x32 (5.625 degree)) achieves RMSE below 0.5 in the first 24 hours.

**Questions:**

1.	Can you follow the settings of WeatherBench2 and re-do the comparison under the 5.625 degree?

---

### Official Review · Reviewer_9yqv · 2025-11-06

**Soundness:** 2
**Presentation:** 2
**Contribution:** 2
**Rating:** 2
**Confidence:** 4

**Summary:**

The authors propose Physics-Aware Tensor Field Neural PDE (PA-TFNP), which uses rotation-equivariant Neural Operators on the sphere. It uses a gradient operator based on spherical transforms and uses boundary conditions.

I think the application of the work is very important but in its current state it is missing too many relevant references as well as SOTA DLWP models to compare to.

**Strengths:**

- Important application of NeuralPDEs for weather
- Nice emphasis on the importance of physical guidance
- Nice use of a hybrid model to model finite differences with ML
- Prediction of several atmospheric variables including geopotential, air temperature and wind speeds.
- Nice ablation study that shows the importance of rotational equivariance
- Nice to highlight that physics inductive biases help the long-term predictions.
- Nice use of ERA5

**Weaknesses:**

- The sentences in the abstract are too long. Please break up into simpler sentences
- SOTA weather models are not compared to, e.g., FourCastNet (Pathak et al., "Fourcastnet: A global data-driven high-resolution weather model using adaptive fourier neural operators", 2022), FourCastNet v3 Bonev et al., https://arxiv.org/pdf/2507.12144, and importantly Bonev et al., "Spherical Fourier Neural Operators: Learning Stable Dynamics on the Sphere", ICML, 2023 which uses a related spherical harmonics basis.
- See Karlbauer et al., "Comparing and contrasting deep learning weather prediction backbones on Navier-Stokes and atmospheric dynamics" for a benchmarking study comparing SOTA DLWP models which should be compared to as well as cited. This work also shows that computing the solutions on a HealPIX mesh can also be beneficial Karlbauer et al., "Advancing Parsimonious Deep Learning Weather Prediction Using the HEALPix Mesh", Journal of Advances in Modeling Earth Systems, 2024.
- Lack of references also in Physics-Informed ML is much broader than PINNs. See Krishnapriyan et al., "Characterizing possible failure modes in physics-informed neural networks", NeurIPS on 2022 on limitations of PINNs and several hard-constrained works, e.g., Negiar et al., "Learning differentiable solvers for systems with hard constraints", ICLR, 2023, Chalapathi et al., "Scaling physics-informed hard constraints with mixture-of-experts", ICLR 2024,  Hansen et al., "Learning Physical Models that Can Respect Conservation Laws", ICML 2024, Mouli et al., "Using uncertainty quantification to characterize and improve out-of-domain learning for pdes", ICML, 2024, Saad et al., "Guiding continuous operator learning through Physics-based boundary constraints", ICLR, 2023, Utkarsh, U., "End-to-End Probabilistic Framework for Learning with Hard Constraints", https://arxiv.org/pdf/2506.07003?, 2025.
- Saad et al., "Guiding continuous operator learning through Physics-based boundary constraints" is important reference for enforcing Neumann BC in Neural Operators.
- The numerical discretization in this work is taken for granted and a simple finite difference scheme is used, which may not be accurate enough for the Navier Stokes equation. In particular, the spatial discretization of PDEs is very important.
-With inputting finite differences, the ML problem simplifies to solving ODEs instead of PDEs.
- The results in Table 1 are mixed with ClimODE sometimes performing better.

**Questions:**

1. Please clarify the novelty of the proposed approach compare to Spherical FNO (SFNO).
2. Please clarify how the approach differes from Method of Lines (MOL) as stated in Section 3 and Neural ODEs.
3. Is the central difference difference method stable? How is the time-step selected?
4. Are other mroe advanced numerical methods compared to, e.g., finite volume and finite elements? Also were upwinding finite difference schemes tried?
5. Do the authors understand hwy ClimODE is performing better on some variables?
6. Do the authors see limitations of the deterministic approaches and future with probabilistic methods?

---

### Note · Authors · 2025-11-19

I have read and agree with the venue's withdrawal policy on behalf of myself and my co-authors.